# Peroxymonosulfate Activation by Fe@N Co-Doped Biochar for the Degradation of Sulfamethoxazole: The Key Role of Pyrrolic N

**DOI:** 10.3390/ijms251910528

**Published:** 2024-09-30

**Authors:** Tong Liu, Chenxuan Li, Xing Chen, Yihan Chen, Kangping Cui, Dejin Wang, Qiang Wei

**Affiliations:** 1School of Resources and Environmental Engineering, Hefei University of Technology, Hefei 230009, China; aimee777@ustc.edu.cn (T.L.); cxli@hfut.edu.cn (C.L.); xingchen@hfut.edu.cn (X.C.); yhchen@hfut.edu.cn (Y.C.); 2Key Laboratory of Nanominerals and Pollution Control of Higher Education Institutes, Hefei University of Technology, Hefei 230009, China; 3School of Resources and Environment, Anqing Normal University, Anqing 246011, China; wangdj@aqnu.edu.cn; 4CAS Key Laboratory of Urban Pollutant Conversion, Department of Environmental Science and Engineering, University of Science and Technology of China, Hefei 230026, China

**Keywords:** biochar, sulfate radicals, peroxymonosulfate, pyrrolic N

## Abstract

In this study, Fe, N co-doped biochar (Fe@N co-doped BC) was synthesized by the carbonization–pyrolysis method and used as a carbocatalyst to activate peroxymonosulfate (PMS) for sulfamethoxazole (SMX) removal. In the Fe@N co-doped BC/PMS system, the degradation efficiency of SMX (10.0 mg·L^−1^) was 90.2% within 40 min under optimal conditions. Radical quenching experiments and electron spin resonance (ESR) analysis suggested that sulfate radicals (SO_4_^•−^), hydroxyl radicals (^•^OH), and singlet oxygen (^1^O_2_) participated in the degradation process. After the reaction, the proportion of pyrrolic N decreased from 57.9% to 27.1%. Pyrrolic N served as an active site to break the inert carbon network structure and promote the generation of reactive oxygen species (ROS). In addition, pyrrolic N showed a stronger interaction with PMS and significantly reduced the activation energy required for the reaction (∆G = 23.54 kcal/mol). The utilization potentiality of Fe@N co-doped BC was systematically evaluated in terms of its reusability and selectivity to organics. Finally, the intermediates of SMX were also detected.

## 1. Introduction

Sulfamethoxazole (SMX), as a common sulfonamide drug, can efficiently treat gastroenteritis, coccidiosis, diarrhea, and other infections and diseases [1]. Due to the high water solubility and potential carcinogenicity of SMX, it has a negative impact on the ecosystem [2,3]. Since SMX is not easy to degrade through conventional treatment processes, it is urgent that we explore an efficient, low-cost, and simple technology for eliminating residual SMX in the aquatic environment.

Advanced oxidation processes (AOPs) have received a huge amount of attention due to their high efficiency and extensive adaptability [4]. Among AOPs, persulfate-based advanced oxidation processes (PS-AOPs) have gradually emerged due to their unique advantages such as a higher oxidation potential and a wider applicability in aquatic environments [5,6]. Due to having a higher oxidation potential (2.5–3.1 V vs. 1.9–2.7 V) and longer half-life (30–40 μs vs. 20 ns) than hydroxyl radicals (^•^OH), sulfate radicals (SO_4_^•−^) can rapidly travel long distances to attack pollutant molecules and oxidize them thoroughly [7]. As a result, there are several approaches to peroxydisulfate (PDS) and peroxymonosulfate (PMS) activation to produce reactive oxygen species (ROS), such as electrolysis, ultrasound, ultraviolet irradiation, heat, and transition metals [8]. Among these activation methods, biochar-based catalysts have the advantages of causing less pollution and showing superior thermal stability [6].

Biochar (BC) is produced via the thermochemical treatment of biomass under an oxygen-limited atmosphere [9]. Biomass feedstock, such as citrus peels, maize stalks, sawdust, corncobs, wood chips, and shrimp shells, is rich and low-cost [7]. Furthermore, BC contains abundant oxygen-containing functional groups (e.g., C-OH, -COOH, C=O) and micropore or mesopore structures [10]. The redox cycle of metal ions (e.g., Fe^3+^/Fe^2+^, Cu^2+^/Cu^+^) can contribute to the generation of ROS in catalytic oxidation processes due to the consecutive cleavage of the peroxide bond (O-O) [11]. Nitrogen doping adjusts the chemical properties of the graphitic structure and increases the electron cloud density of the surrounding carbon atoms, and, thus, regulates the chemical properties of biochar-based catalysts [12].

Herein, Fe, N co-doped biochar (Fe@N co-doped BC) was prepared through the carbonization–pyrolysis method and was used as a PMS activator for the elimination of SMX. Characterizations and batch experiments were employed to assess the degradation ability of Fe@N co-doped BC. Then, the active species in the Fe@N co-doped BC/PMS system were identified by quenching tests and electron spin resonance (ESR) tests. The activation mechanism of PMS at different N configurations was uncovered through X-ray photoelectron spectroscopy (XPS) analysis and a density functional theory (DFT) calculation. Additionally, its reusability, universal applicability, and mineralization capacity were evaluated.

## 2. Results and Discussion

### 2.1. Characterization

The morphologies of samples were explored using scanning electron microscopy (SEM). The surface of the primary BC is smooth and blocky. After the doping of urea, a layer of bright blocky stack appeared on the surface of the N-doped BC (Figure 1a,b). A large number of iron oxide particles are gathered on the surface of the Fe-doped BC, while the carbon substrate exhibits a regular blocky structure (Figure 1c). The surface of Fe@N co-doped BC is loaded with tiny iron oxide nanoparticles, and no isolated distribution of nanoparticles is observed outside the carbon matrix, indicating that these nanoparticles have been anchored to the carbon layer (Figure 1d).

In addition, the ultrastructure of the crystal lattice of the Fe@N co-doped BC was studied with a high-resolution transmission electron microscope (HRTEM) and a selected area electron diffraction (SAED) pattern. As shown in Figure 2a, iron oxide particles are successfully grown on the surface of BC by the carbonization–pyrolysis method, and these iron oxide particles can be used as the active site for activating PMS. The (444) crystal faces belonging to Fe_3_O_4_ with a lattice spacing of 1.21 nm were calculated by Fast Fourier Transform (FFT), as depicted in Figure 2b. The Miller concentric rings also exhibit an Fe@N co-doped BC crystal phase structure, and the Miller index of (111) and (311) also demonstrates the formation of Fe_3_O_4_ on the surface of the BC (Figure 2c). Energy-dispersive spectroscopy (EDS) shows the distribution of five elements, C, Fe, Si, N, and O, and also proves that iron and nitrogen have been successfully incorporated into the carbon material (Figure 2d).

XRD spectra of samples are depicted in Figure 3a. The diffraction peaks of BC appear at about 21.6°, corresponding to the C (111) crystal plane. Interestingly, N-doped BC significantly increases the diffraction peak width at 26.6°, which may be due to the incorporation of urea. It has been reported that the addition of urea can improve the graphitization degree of biochar [12]. Fe@N co-doped BC and Fe-doped BC exhibit distinct diffraction peaks at 30.1°, 35.4°, 43.1°, 53.4°, 56.9°, and 62.5°, corresponding to the crystal plane of (220), (311), (400), (422), (511), and (440), respectively. It was demonstrated that Fe_3_O_4_ nanoparticles had been successfully loaded onto biochar surfaces (JCPDS 89-2355).

The FT-IR spectrum shows the surface functional groups (Figure 3b). The peaks at 3446 cm^−1^ are attributed to the stretching vibration of -OH groups [13]. The content of -OH groups on the surface of Fe@N co-doped BC is significantly higher than that on Fe-doped BC, N-doped BC, and BC. The absorption peaks at 2922 cm^−1^ and 2852 cm^−1^ are attributed to -CH_2_ and -CH_3_ groups, respectively [14]. The absorption peaks at 1116 cm^−1^ and 1384 cm^−1^ are attributed to the vibrations of C-OH and C-C [15]. In addition, the absorption peak at 570 cm^−1^ is caused by the tensile vibration of Fe-O, which also indicates that Fe and N co-doping breaks the inert structure of the carbon network and generates a new active center [16].

N_2_ adsorption–desorption isotherms (Figure 3c) indicate that the specific surface areas (SSAs) of N-doped BC (287.67 m^2^·g^−1^) are larger than those of BC (194.86 m^2^·g^−1^), Fe@N co-doped BC (269.21 m^2^·g^−1^), and Fe-doped BC (216.38 m^2^·g^−1^) (Table 1).

As shown in Figure 3d, the appearance of both the D-band and G-band of catalysts reveals the co-presence of disordered and crystalline graphite structures [17]. Furthermore, the ratio of I_D_/I_G_ is the key parameter in indicating the defective degree of catalysts [18,19]. The I_D_/I_G_ is 1.07, 0.81, 1.02, and 2.14 for BC, Fe-doped BC, N-doped BC, and Fe@N co-doped BC, respectively. These results demonstrate that Fe@N co-doped BC has obtained abundant defects (vacancy and edge defects) during pyrolysis, which is conducive to catalytic oxidation [10].

### 2.2. Catalytic Oxidation of SMX

The adsorption efficiency of different catalysts for SMX is shown in Figure 4. Under the conditions of a pH of 7.0, a catalyst dosage of 0.4 g/L, and a reaction temperature of 25 °C, the adsorption efficiency of BC, Fe-doped BC, N-doped BC, and Fe@N co-doped BC for SMX is 25.8%, 23.2%, 30.0%, and 27.9%, respectively, for 40 min. It is clear that the adsorption performance of N-doped BC is higher than that of BC, Fe-doped BC, and Fe@N co-doped BC, which may be due to the doped urea, which leads to a porous structure [20].

Without adding any catalyst, the degradation efficiency of SMX in sole PMS system is 8.0%, which indicates that the self-decomposition ability of PMS is low (Figure 5a). In the BC/PMS system, the degradation efficiency of SMX increased to 67.3%, indicating that BC can activate PMS to a certain extent. N and Fe doping alone can partially improve the degradation efficiency of BC, and the degradation efficiency of N-doped BC and Fe-doped BC for SMX is 69.0% and 71.0%, respectively. Fe@N co-doped BC has the highest catalytic activity under the conditions of a pH of 7.0 and an initial SMX concentration of 10.0 mg/L, an Fe@N doped BC of 0.4 g/L, a PMS concentration of 0.6 mM, and a reaction temperature of 25 °C; the degradation efficiency of SMX can reach 90.2%.
(1)$$In(C_{t}/C_{0})=-kt$$

In addition, the pseudo-first-order kinetic model was used to study the degradation process of SMX in different catalytic systems (Equation (1)) (Figure 5b). The linear regression coefficients (R^2^) were all over 0.90, indicating that the reaction kinetics of degradation of SMX by each catalyst were consistent with pseudo-first-order kinetics. The reaction rate constant (*k*_obs_) of Fe@N co-doped BC/PMS system was 0.041 min^−1^, being about 1.50, 1.40, and 1.60 times that of N-doped BC/PMS, Fe-doped BC/PMS, and BC/PMS systems, respectively (Appendix A). The typical pollutant degradation efficiencies in different systems are shown in Appendix A.

To explore the catalytic performance of Fe@N co-doped BC in depth, the residual PMS concentration was detected by the ABTs colorimetric method. The decomposition efficiency of PMS in the Fe@N co-doped BC/PMS system was 98.6% within 40 min (Figure 6). It was suggested that the favorable degradation efficiency of SMX in the Fe@N co-doped BC /PMS system could be due to the rapid decomposition of PMS.

The effects of Fe@N co-doped BC dosage, PMS concentration, initial pH, and SMX concentration on SMX degradation were investigated (Figure 7). In the tested range, an increase in Fe@N co-doped BC dosage and PMS concentration results in an increase in the SMX degradation efficiency, as an increase in the Fe@N co-doped BC and PMS dosage is conducive to the rapid generation of ROS in the Fe@N co-doped BC/PMS system [21]. The initial pH of the solution had a significant effect on the SMX removal efficiency. Under neutral conditions (pH = 7.0), the degradation efficiency of SMX in the Fe@N co-doped BC/PMS system was as high as 90.2%. Under the condition of acidity (pH = 3.0), the degradation efficiency of SMX decreased from 90.2% to 57.6%. Under acidic conditions, the increase in iron leaching is expected to lead to a lower catalytic activity in Fe@N co-doped BC, with fewer active sites [22]. Under alkaline conditions (pH = 11.0), the degradation efficiency of SMX decreased from 90.2% to 65.5%. When the initial concentrations of SMX were 0.5 mg/L, 1.0 mg/L, 5.0 mg/L, and 10.0 mg/L, the degradation efficiencies of SMX were 98.8%, 98.6%, 95.7%, and 90.2%, respectively. In the test range, the initial concentration of pollutants had a slight effect on the degradation efficiency of SMX.

Co-existing ions can restrain SMX removal by interacting with reactive species [23]. Typical anions (Cl^−^, H_2_PO_4_^−^, HCO_3_^−^), and humic acid (HA) restrained the SMX elimination to varying degrees (Appendix A). When 5.0 mM and 500.0 mM Cl^−^ were present in the Fe@N co-doped BC/PMS system, the SMX removal efficiency decreased from 90.2% to 79.0% and 70.0%, respectively, within 40 min. The corresponding *k*_obs_ decreased from 0.041 min^−1^ to 0.025 min^−1^ and 0.021 min^−1^, which was due to the generation of Cl^•^, Cl_2_, and HOCl (Equations (2)–(6)) [24,25]. HCO_3_^−^ also have a quenching effect on reactive radicals (Equations (7) and (8)). Compared with Cl^−^, HCO_3_^−^ and H_2_PO_4_^−^ have a more significant inhibitory effect on SMX degradation due to the fact that HCO_3_^−^ reacts with ^•^OH and SO_4_^•−^ to form CO_3_^•−^, which has a lower redox potential [26]. Similarly, H_2_PO_4_^−^ reacts with SO_4_^•−^ to produce less active free radicals, thereby reducing the degradation efficiency of SMX. In addition, H_2_PO_4_^−^ easily binds to the active sites on the surface of Fe@N co-doped BC, preventing the catalyst from producing enough ROS to degrade SMX [27]. When 20.0 mg/L and 50.0 mg/L HA were present in the reaction system, the degradation efficiencies of SMX decreased from 90.2% to 81.0% and 62.0%, respectively, and the corresponding *k*_obs_ decreased from 0.041 min^−1^ to 0.033 min^−1^ and 0.019 min^−1^, respectively, within 40 min. On one hand, HA can consume ^•^OH and SO_4_^•−^; on the other hand, HA has a strong π-π superposition effect, which is easy to deposit on the surface of Fe@N co-doped BC, thus inhibiting the interaction between Fe@N co-doped BC, PMS and SMX [28].
(2)$$Cl^{-}+SO_{4}^{-}\to SO_{4}^{2-}+Cl$$
(3)$$HSO_{5}^{-}+Cl^{-}\to SO_{4}^{2-}+HOCl$$
(4)$$HSO_{5}^{-}+2Cl^{-}+H^{+}\to SO_{4}^{2-}+{Cl}_{2}+H_{2}O$$
(5)Cl−+OH·→HOCl−
(6)$$HOCl^{\cdot -}+H^{+}\to {Cl}^{\cdot }+H_{2}O$$
(7)HCO3−+OH·→H2O+CO32−
(8)$$HCO_{3}^{-}+SO_{4}^{\cdot -}\to SO_{4}^{2-}+HCO_{3}^{\cdot }$$

Figure 8a shows the degradation efficiencies of various pharmaceuticals and personal care products (PPCPs) in Fe@N co-doped BC/PMS system. The degradation efficiencies of tetracycline hydrochloride (TC), 4-chloro-3-methyl phenol (CMP), norfloxacin (NOR), and acetaminophen (ACT) were 95.8%, 93.3%, 91.5%, and 94.2%, respectively, in 40 min. In general, representative PPCPs could be removed efficiently in an Fe@N co-doped BC/PMS system. To study the reusability of Fe@N co-doped BC, five consecutive degradation tests were conducted (Figure 8b). After five cycles, the degradation efficiency of SMX within 40 min decreased from 90.2% to 63.0%. This could be due to the sedimentation of intermediates on the surface of Fe@N co-doped BC, which occupied the reactive sites for activating PMS. The saturation magnetization value of Fe@N co-doped BC was 17.43 emu/g (Figure 8c). Fe@N co-doped BC exhibited a strong magnetic response to an external magnetic field and was easily separated from the SMX solution, facilitating the recycling of the catalyst. Figure 8d) displayed the continuous leaching of Fe species in the redox process. Fortunately, less than 1.04 mg/L of Fe species was leached in the reaction solution.

### 2.3. Mechanism Discussion

#### 2.3.1. Identification of ROS

The above results indicate that SMX can be degraded rapidly in the Fe@N co-doped BC/PMS system, while previous studies have shown that SMX may be degraded via either a radical pathway (O_2_^•−^, ^•^OH, or SO_4_^•−^) or a non-radical pathway (^1^O_2_ or biochar-PMS* complex) in a heterogeneous catalytic oxidation system [6]. To determine the active species produced in this system and their contribution to SMX degradation, quenching experiments and ESR tests were performed. ^•^OH, ^•^OH/SO_4_^•−^, and O_2_^•−^ were quenched by *tert*-butanol (TBA), methanol (MeOH), and *p*-benzoquinone (*p*-BQ), respectively, while furfuryl alcohol (FFA) and *L*-histidine (*L*-his) were used as trapping agents for ^1^O_2_ [29,30]. The SMX removal efficiencies decreased from 92.0% to 33.4%, 73.5%, 86.7%, 73.2%, and 71.8% within 40 min after adding MeOH (0.5 M), TBA (0.5 M), *p*-BQ (0.1 M), *L*-his (0.1 M), and FFA (0.1 M), respectively (Figure 9). These results indicate that ^•^OH and SO_4_^•−^ exist in the system and SO_4_^•−^ may play a dominant role. Since *p*-BQ has a slight inhibitory effect on SMX degradation, the system does not produce large amounts of O_2_^•−^. The contribution of each reactive species is listed in Appendix A.

ESR was performed to further prove the presence of ROS in the Fe@N co-doped BC/PMS system using 2,2,6,6-tetramethyl-4-piperidinol (TEMP) and 5,5-dimethyl-1-pyrroline N-oxide (DMPO) as spin-trapping agents [31,32]. A spectrum with seven main peaks belonging to DMPO-X was observed for the Fe@N co-doped BC/PMS system, indicating that Fe@N co-doped BC activated PMS to produce ^•^OH and SO_4_^•−^ (Figure 10a). The signal intensity in the Fe@N co-doped BC/PMS system was significantly higher than that of the Fe-doped/PMS and N-doped/PMS system, indicating that there is a synergistic effect between Fe and N on the surface of biochar. When Fe@N co-doped BC, PMS, and TEMP were added to the solution, a typical triplet signal with the intensity ratio of 1:1:1 verified the existence of ^1^O_2_ (Figure 10b). The self-decomposition of PMS could produce a small amount of ^1^O_2_ [33]. In summary, ^•^OH, SO_4_^•−^, and ^1^O_2_ are produced in this system, in which SO_4_^•−^ is the main active species and leads the degradation process of the free radical pathway, while ^1^O_2_ leads the degradation process of the non-free radical pathway.

#### 2.3.2. Reaction Mechanism

The above experiments demonstrate the production of ^•^OH, SO_4_^•−^, and ^1^O_2_ in the Fe@N co-doped BC/PMS system, and on the basis of this, the general catalytic mechanism is proposed. First, ^•^OH and SO_4_^•−^ may be produced by the destruction of the O-O bond of the PMS by free-flowing π electrons on the *sp*^2^ hybrid carbon (Equations (1)–(3)). As can be seen from the XPS spectrum (Figure 11a), the surface of Fe@N co-doped BC is successfully doped with iron and nitrogen, and the specific atomic percentages of the obtained catalyst are shown in Appendix A. Fe@N co-doped BC surface contains C-OH, COOH, and C=O groups, which can serve as an active site (Figure 11b). As an electron donor functional group, C-OH is more inclined to activate PMS to produce free radicals, such as ^•^OH and SO_4_^•−^. As electron withdrawing functional groups, C=O and COOH are more inclined to degrade SMX through the non-radical pathway. After the catalytic reaction, the Fe(II) content decreased from 43.7% to 40.8%, while the Fe(III) content increased from 36.4% to 37.9% (Figure 11c). The doped Fe(II) reacts with HSO_5_^−^ to form ^•^OH and SO_4_^•−^ (Equations (12) and (13)), and the generated free radicals can be converted flexibly. Subsequently, Fe(III) reacts with HSO_5_^−^ to produce SO_5_^•−^, and Fe(III) is reduced to Fe(II) and maintains a complete cycle (Equation (14)). These results indicate that Fe_3_O_4_ plays a leading role in the activation of PMS in the free radical pathway.

It has been reported that carbon-based materials can also activate PMS to degrade organic micropollutants through non-free radical pathways, and the non-free radical mechanism may be related to nitrogen doping [34,35]. N 1s can be divided into three peaks, including pyrrolic N, pyridinic N, and graphitic N (Figure 11d). The content of graphitic N and pyridinic N increased from 17.6% and 24.5% to 42.4% and 30.5%, respectively, while the content of pyrrolic N decreased from 57.9% to 27.1%. Therefore, pyrrolic N is considered to be the main active species in the redox process.
(9)$$\pi -electrons+HSO_{5}^{-}\to SO_{4}^{\cdot -}+OH^{-}$$
(10)π−electrons+HSO5−→OH·+SO42−
(11)$$SO_{4}^{\cdot -}+H_{2}O\to H^{+}+SO_{4}^{2-}+OH$$
(12)$$Fe^{2+}+HSO_{5}^{-}\to Fe^{3+}+SO_{4}^{\cdot -}+OH^{-}$$
(13)$$Fe^{2+}+HSO_{5}^{-}\to Fe^{3+}+SO_{4}^{2-}+\cdot OH$$
(14)$$Fe^{3+}+HSO_{5}^{-}\to Fe^{2+}+SO_{5}^{\cdot -}+H^{+}$$

The PMS activation performances of three N configurations, pyrrolic N, pyridinic N, and graphitic N, were compared, as shown in Figure 12. From the top view, it can be seen that PMS can be stably adsorbed on top of pyrrolic N, pyridinic N, and graphitic N. For the reactant structure, the distance (D) between PMS and pyrrolic N is the shortest (2.986 Å). The transition state energy barrier of pyrrolic N/PMS, pyridinic N/PMS, and graphitic N/PMS was calculated by the B3LYP/6-31G (d,p) method as 23.54 kcal/mol, 27.81 kcal/mol, and 26.45 kcal/mol, respectively. The above results showed that pyrrolic N could significantly reduce the activation energy required for the reaction, and the catalytic activity of this active site was the highest.

To investigate the charge transfer efficiency on the surface of carbon-based materials, electrochemical tests were carried out. Electrochemical impedance spectroscopy (EIS) shows that Fe@N co-doped BC has a smaller semicircle diameter and therefore lower electron migration resistance than that of BC, Fe-doped BC, and N-doped BC (Figure 13). The hydrophilicity of carbon-based materials was further investigated by measuring the contact angle [20] (Figure 14). The contact angle of Fe@N co-doped BC is 4.7°, which is lower than that of BC (14.8°), N-doped BC (9.0°), and Fe-doped BC (9.3°), which means that Fe@N co-doped BC has higher hydrophilicity, and it is easier to obtain excellent degradation properties using this option.

### 2.4. Degradation Pathways of SMX

The intermediate products of SMX degradation were investigated by UPLC-TOF/MS (Figure 15 and Appendix A). In pathway I, the C atom on the isoxazole ring of SMX undergoes an electrophilic addition reaction to produce the product I (*m*/*z* = 288) [1]. Subsequently, the S-N bond of SMX breaks and product I is transformed into product II (*m*/*z* = 133) and product III (*m*/*z* = 190). Under ROS attack, the N-O bond on the isoxazole ring breaks to produce product IV (*m*/*z* = 117), which is eventually mineralized into CO_2_ and H_2_O. At the same time, a ring-opening reaction may occur on the isoxazole ring of product I to form product VI (*m*/*z* = 246), as well as a subsequent dehydrogenation reaction to form intermediate product VII (*m*/*z* = 216). In pathway III, SMX undergoes electrophilic substitution to produce product IX (*m*/*z* = 270). Subsequently, the S-N bond of SMX breaks and product IX is transformed into product X (*m*/*z* = 190) and product XI (*m*/*z* = 99). Under ROS attack, product XI is further oxidized to product XII and product XIII. In summary, the degradation mechanism of SMX in the Fe@N co-doped BC/PMS system is shown in Figure 16.

## 3. Materials and Methods

### 3.1. Preparation of Catalysts

Slow pyrolysis, as a typical carbonization method, is commonly used for the conversion of biomass wastes into biochar with high-carbon and low-volatile-matter contents [36]. In this work, Fe@N co-doped BC was fabricated using a carbonization–pyrolysis strategy, which has been proved to be simple and reliable [12]. Rice husk was purchased from a biomass pyrolysis power plant in China. Firstly, biomass was washed, dried in a vacuum drying oven at 100 °C for 12 h, ground, and then passed through a 100-mesh sieve to acquire thin powders for further use. Then, 5.56 g FeSO_4_·7H_2_O, 3.00 g urea, and 2.50 g ascorbic acid was dissolved into 100 mL deionized water, after which 3.00 g powders were added into the solution and adequately shaken in a thermostatic shaking bath. The mixture was dried in a vacuum-drying oven for 24 h and calcined at a constant calcination temperature (800 °C) under a N_2_ atmosphere (5 °C/min. of heating rate) for 2 h. The catalyst was washed with deionized water several times, and the resulting composites were denoted as Fe@N co-doped BC. For comparison, Fe-doped BC, N-doped BC, and BC were synthesized using the same method.

### 3.2. Reaction Procedures

Batch experiments were carried out in 50 mL vials containing 50 mL of SMX to evaluate the catalytic ability of Fe@N co-doped BC on PMS activation. The range of Fe@N co-doped BC dosing was 0.2–0.8 g/L, the concentration of PMS was 0.2–0.8 mM, the pH was 3.0–11.0, the SMX concentration was 0.5–10.0 mg/L, and the reaction temperature was set to 25 °C. All experiments were carried out at 150 rpm, and 1.0 mL reaction solution was taken periodically at intervals and filtered through a 0.22 μm filter. The excess reaction was terminated by adding 1 mL Na_2_S_2_O_3_ (0.2 M). Before the reaction, 0.1 M NaOH or H_2_SO_4_ was added into the reaction solution to adjust the initial pH values. All experiments were performed in triplicate.

## 4. Conclusions

In summary, Fe@N co-doped BC was prepared by the carbonization–pyrolysis method, and the catalyst obtained abundant catalytic centers (Fe_3_O_4_, pyrrolic N) and defect sites (I_D_/I_G_ = 0.97). Under optimal conditions, the degradation efficiency of SMX in the Fe@N co-doped BC/PMS system is 90.2%. The degradation process conforms to pseudo-first-order kinetics with a *k*_obs_ of 0.041 min^−1^. In addition, various typical PPCPs could be efficiently degraded in the Fe@N co-doped BC/PMS system within 40 min. The quenching experiment and ESR test results show that ^•^OH, SO_4_^•−^, and ^1^O_2_ were produced in the Fe@N co-doped BC/PMS system, in which SO_4_^•−^ was the main active species. SO_4_^•−^ can be produced in the redox cycle between Fe(III) and Fe(II), and Fe_3_O_4_ acts as an important catalytic center to break the inert carbon network structure and promote the formation of ROS in the reaction system. Before and after the reaction, the content of pyrrolic N decreased from 57.9% to 27.1%. As the main active site, pyrrolic N showed stronger interaction with PMS and significantly reduced the activation energy required for the reaction (∆G = 23.54 kcal/mol). Therefore, this work first solves the problem of low catalytic activity caused by the lack of catalytic center in carbon materials. Moreover, the excellent reusability, oxidation ability, stability, good mineralization ability, and low cost confirm that Fe@N co-doped biochar is a promising catalyst in industrial wastewater advanced treatment.

## Figures and Tables

**Figure 1 ijms-25-10528-f001:**
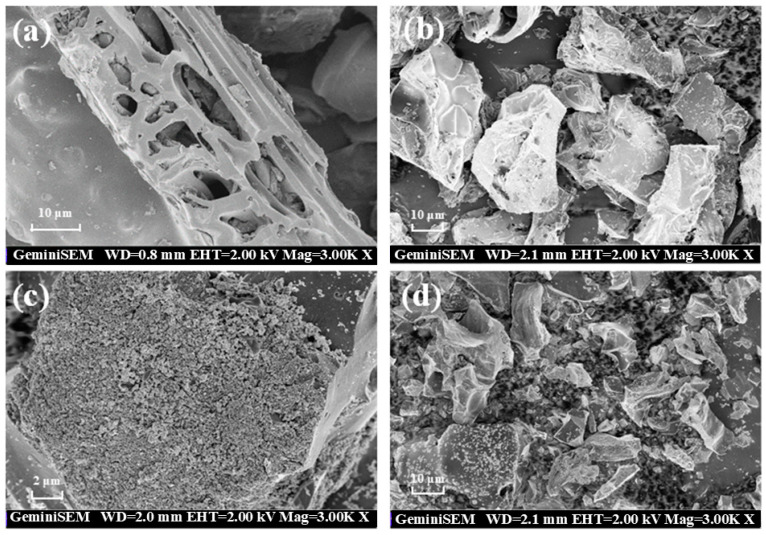
SEM images of BC (**a**), N-doped BC (**b**), Fe-doped BC (**c**), and Fe@N co-doped BC (**d**).

**Figure 2 ijms-25-10528-f002:**
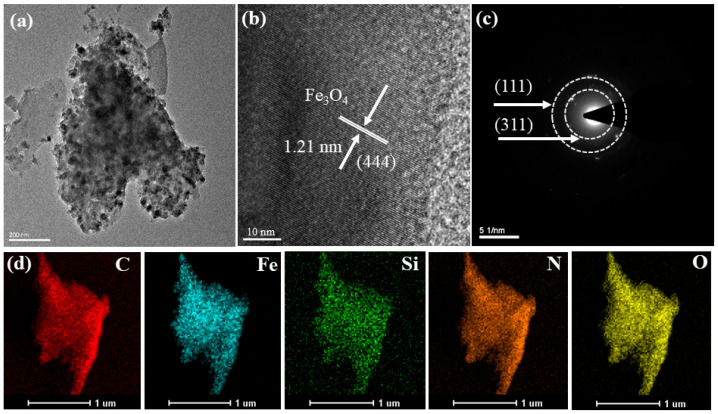
HRTEM images (**a**,**b**), SAED (**c**), and element distribution (**d**) of Fe@N co-doped BC.

**Figure 3 ijms-25-10528-f003:**
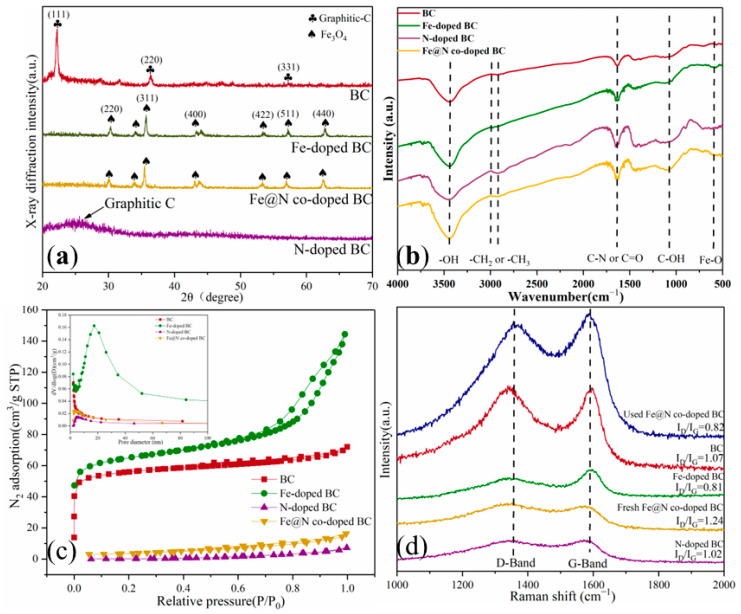
XRD patterns (**a**), FT-IR spectra (**b**), nitrogen adsorption–desorption isotherms (**c**), and Raman spectra (**d**) of catalysts.

**Figure 4 ijms-25-10528-f004:**
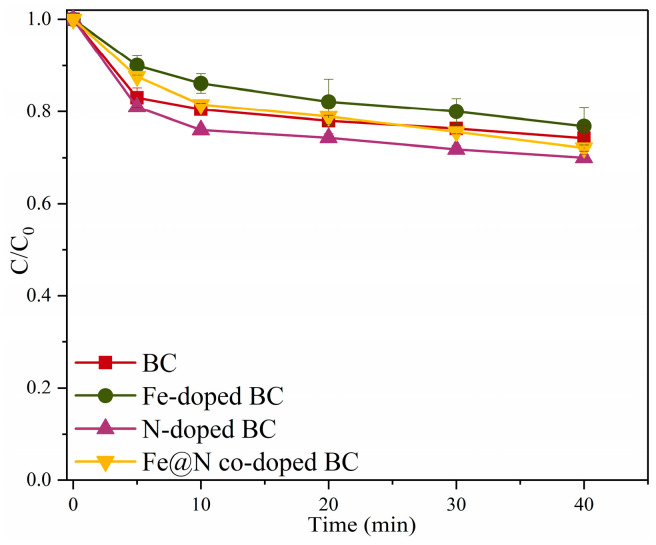
The adsorption efficiencies of SMX in various reaction systems. Reaction conditions: [SMX]_0_ = 10.0 mg/L, [catalyst]_0_ = 0.4 g/L, initial pH = 7.0, and T = 25 °C.

**Figure 5 ijms-25-10528-f005:**
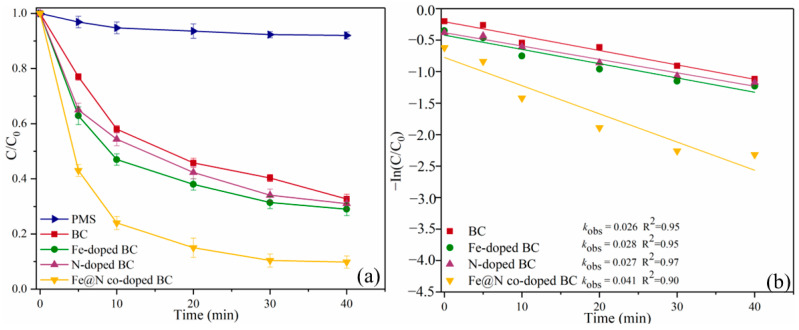
SMX degradation efficiencies (**a**), and *k*_obs_ in different systems (**b**). Reaction conditions: [SMX]_0_ = 10.0 mg/L, [catalyst]_0_ = 0.4 g/L, [PMS]_0_ = 0.6 mM, initial pH = 7.0, and T = 25 °C.

**Figure 6 ijms-25-10528-f006:**
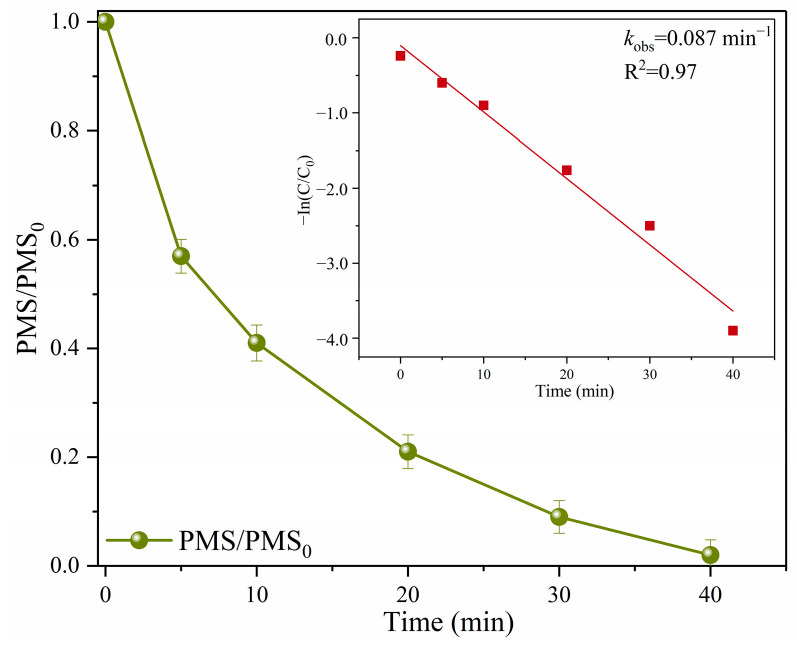
Residual PMS in the Fe@N co-doped BC/PMS system. Reaction conditions: [SMX]_0_ = 10.0 mg/L, [catalyst]_0_ = 0.4 g/L, [PMS]_0_ = 0.6 mM, initial pH = 7.0, and T = 25 °C.

**Figure 7 ijms-25-10528-f007:**
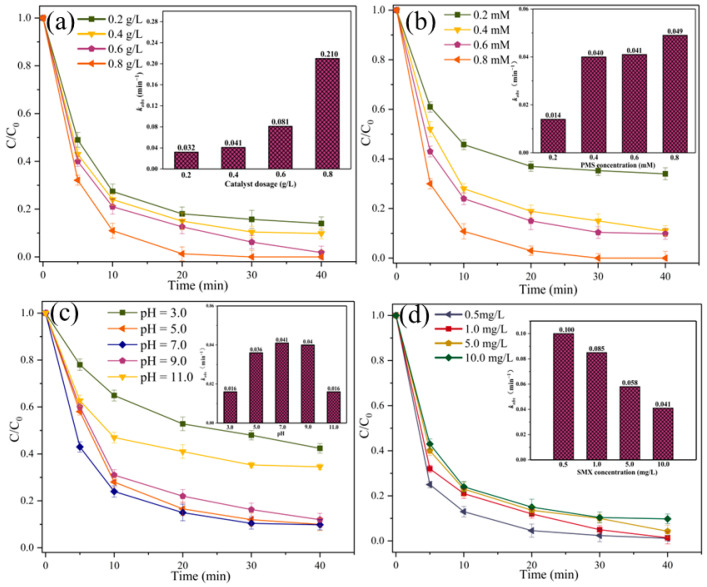
Effects of different parameters on SMX removal in the Fe@N co-doped BC/PMS system: (**a**) catalyst dosage, (**b**) PMS concentration, (**c**) solution pH, (**d**) SMX concentration (conditions: pH = 7.0; [SMX]_0_ = 10.0 mg/L; [catalyst] = 0.4 g/L; [PMS] = 0.6 mM; and T = 25 °C).

**Figure 8 ijms-25-10528-f008:**
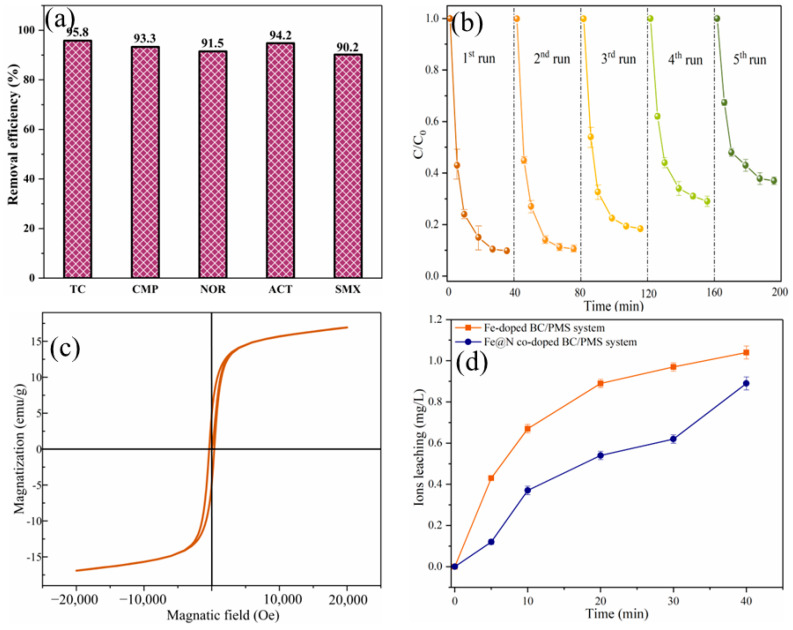
The removal of various pollutants in the Fe@N co-doped BC/PMS system (**a**), reusability of Fe@N co-doped BC after five consecutive cycles (**b**), the room-temperature magnetization curve of Fe@N co-doped BC (**c**), and the concentration of leached ions (**d**) (conditions: [substrate]_0_ = 10.0 mg/L; [catalyst]_0_ = 0.4 g/L; [PMS]_0_ = 0.6 mM; pH = 7.0; T = 25 °C).

**Figure 9 ijms-25-10528-f009:**
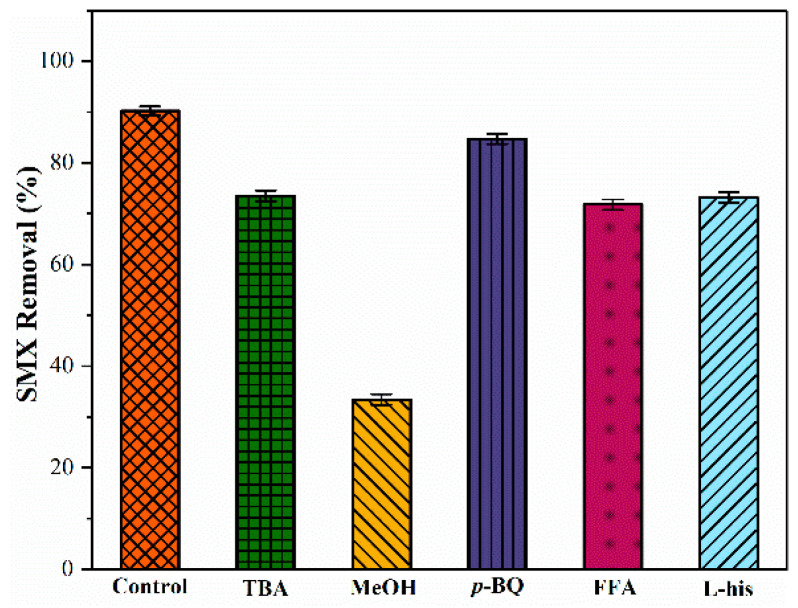
Quenching tests for the reactive species (conditions: pH = 7.0; [SMX]_0_ = 10.0 mg/L; [catalyst] = 0.4 g/L; [PMS] = 0.6 mM; T = 25 °C; [MeOH] = [TBA] = 500.0 mM; and [FFA] = [*L*-his] = [*p*-BQ] = 100.0 mM).

**Figure 10 ijms-25-10528-f010:**
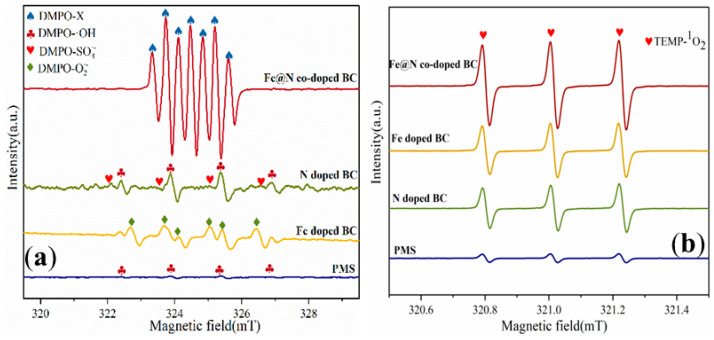
ESR spectra of (**a**) DMPO-X and (**b**) TEMP-^1^O_2_ (conditions: [SMX]_0_ = 10.0 mg/L; [catalyst]_0_ = 0.4 g/L; [PMS]_0_ = 0.6 mM; pH = 7.0; T = 25 °C; and [TEMP] = [DMPO] = 10.0 mM).

**Figure 11 ijms-25-10528-f011:**
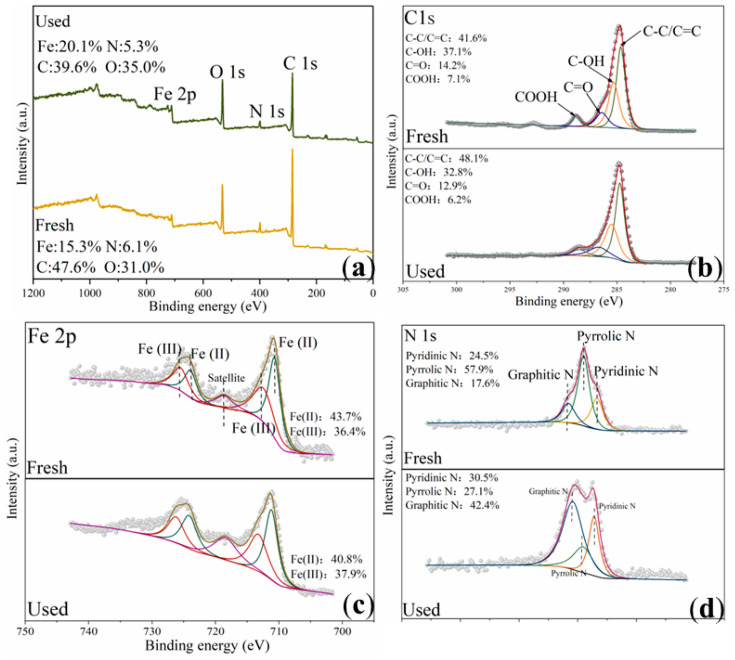
XPS spectra of the full-range survey (**a**), C 1s (**b**), Fe 2p (**c**), and N 1s (**d**) for Fe@N co-doped BC.

**Figure 12 ijms-25-10528-f012:**
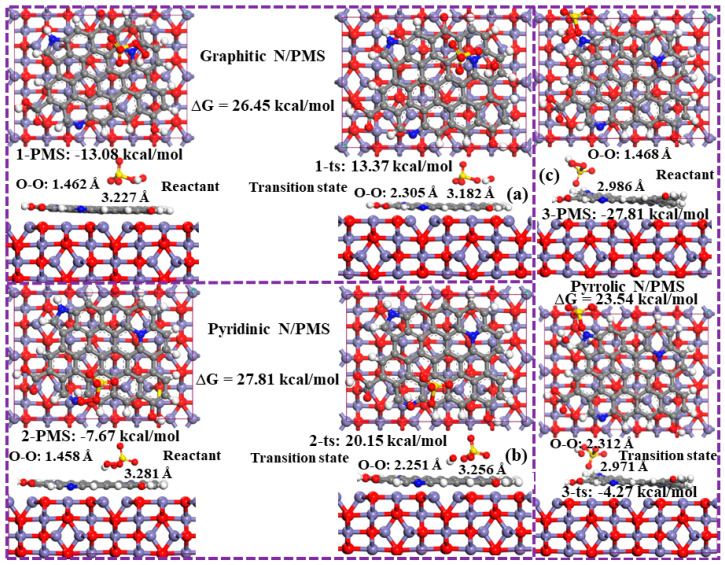
The optimization structures of PMS adsorption on different N substitution sites and the corresponding transition state. (**a**) Graphitic N/PMS, (**b**) pyridinic N/PMS, and (**c**) pyrrolic N/PMS.

**Figure 13 ijms-25-10528-f013:**
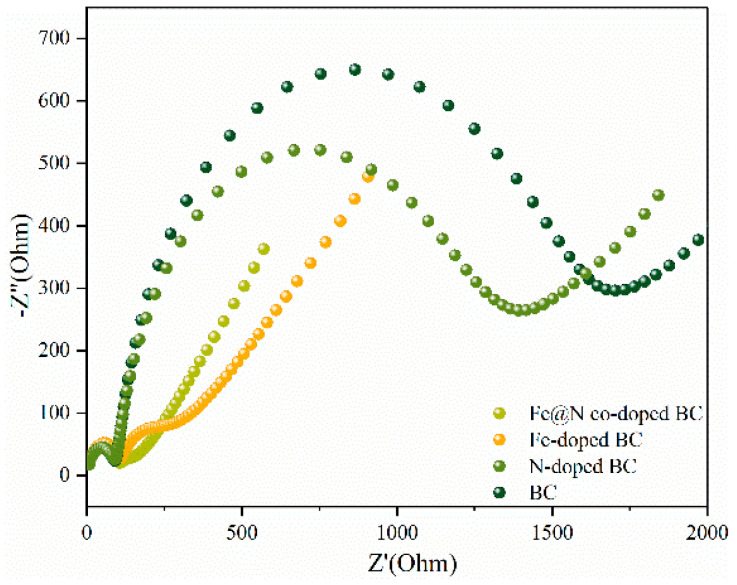
EIS Nyquist plots of various carbon-based catalysts.

**Figure 14 ijms-25-10528-f014:**
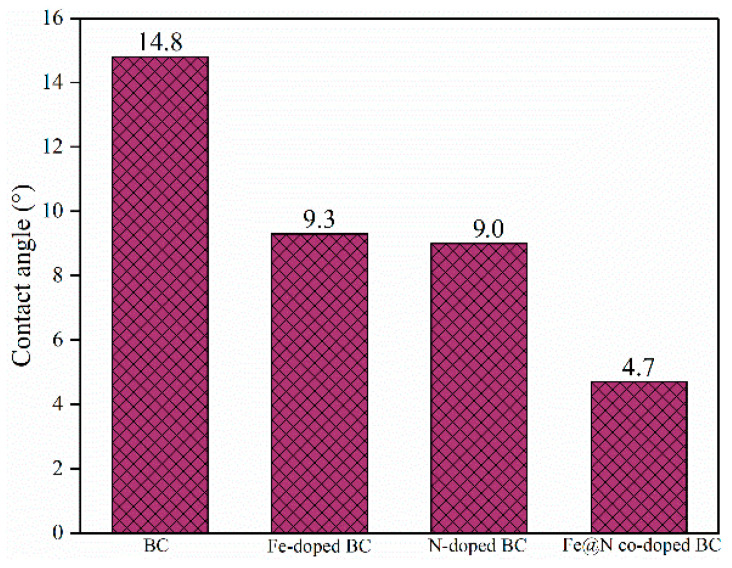
The contact angle of catalysts.

**Figure 15 ijms-25-10528-f015:**
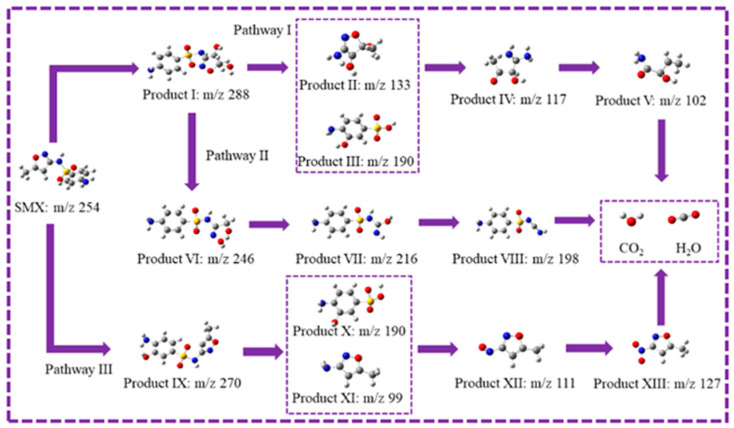
Possible pathways for the degradation of SMX in the Fe@N co-doped BC/PMS system.

**Figure 16 ijms-25-10528-f016:**
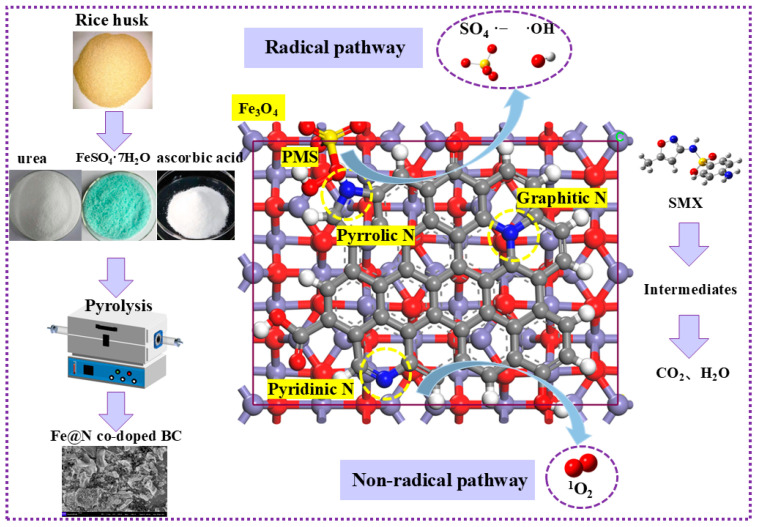
Proposed mechanism of SMX degradation in the Fe@N co-doped BC/PMS system.

**Table 1 ijms-25-10528-t001:** BET and BJH results for different catalysts.

Samples	S_BET_ (m^2^/g) ^a^	Pore Volume (cm^3^/g) ^a^	Average Pore Diameter (nm) ^a^	I_D_/I_G_ ^b^
BC	194.86	0.110	2.46	1.07
Fe-doped BC	216.38	0.220	3.64	0.81
N-doped BC	287.67	0.300	3.56	1.02
Fe@N co-doped BC	269.21	0.240	3.48	2.14

^a^ Obtained by Langmuir modeling studies. ^b^ Analyzed by Raman spectra.

## Data Availability

The data that support the findings of this study are available from the corresponding author upon reasonable request.

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
