# Peer review of "Peroxymonosulfate Activation by Fe@N Co-Doped Biochar for the Degradation of Sulfamethoxazole: The Key Role of Pyrrolic N"

_ijms, 2024, doi:10.3390/ijms251910528_

Round 1
Reviewer 1 Report
Comments and Suggestions for Authors
The manuscript entitled “Peroxymonosulfate activation by Fe@N co-doped BC for the degradation of sulfamethoxazole: The key role of pyrrolitic N” is a very interesting work.
The research program and the way of presenting the results are very good. The information contained in the text is clear. Both the description of the catalyst, its efficiency and the mechanism of its action are exemplary. However, before publication the authors need to correct minor errors and clarify several issues.
1. The unit of the reaction rate is not %. In most cases, instead of the term “rate” it should be “efficiency” or “yeld”
2. The equipment manufacturer is “USA” and not “America”
3. The adsorption efficiency in all cases is similar. The sentence “which indicates that the iron species loaded on the carbon surface may block part of the adsorption sites, then the adsorption efficiency of the material is reduced” is an overinterpretation.
4. The t0 points in Figures 5A and 5B (for Fe@N co-doped BC) do not have corresponding values. This may affect the interpretation of the results and conclusions about the type of kinetics.
5. Please explain the low activity of the catalyst at pH 3.
6. Please do not use the term "strong acidity" at pH 3. "acidity" is enough
7. I suggest moving Figures S2 and S4 from the supplement to the text.
8. The results in Table S4 do not correlate with the information on the elution of Fe ions. Please explain this.
9. The formulas in Figure S4 will be more legible as structural patterns. The current form is an unnecessary eccentricity.
10. Figure 11 in its current form is difficult to read.
11. In the first paragraph (3.2), please do not provide the SMX concentration.
12. For what purpose did the authors use the TOC analyzer?
13. The material used to synthesize the catalyst (BC) may be different. Please provide its elemental composition?
Author Response
Dear Editor,
We would like to thank you very much for your time and consideration on our manuscript.
According to the reviewers’ comments, we have carefully revised the manuscript, which we hope meet with their approval. The following responses are inserted with each issue raised by the reviewers. The revisions are marked in red color in the revised manuscript.
Detailed responses to reviews’ comments
Reviewer #1
Dear Editor and Authors:
The manuscript entitled “Peroxymonosulfate activation by Fe@N co-doped BC for the degradation of sulfamethoxazole: The key role of pyrrolitic N” is a very interesting work.
The research program and the way of presenting the results are very good. The information contained in the text is clear. Both the description of the catalyst, its efficiency and the mechanism of its action are exemplary. However, before publication the authors need to correct minor errors and clarify several issues.
Comments 1: The unit of the reaction rate is not %. In most cases, instead of the term “rate” it should be “efficiency” or “yeld”.
Response 1: Thank you very much for your careful review of our work. Based on your opinion, we have replaced all the words "rate" with "efficiency" (revised in manuscript).
Comments 2: The equipment manufacturer is “USA” and not “America”.
Response 2: Thanks for your careful review. Based on your opinion, we have corrected this error (revised in supplementary material).
Comments 3: The adsorption efficiency in all cases is similar. The sentence “which indicates that the iron species loaded on the carbon surface may block part of the adsorption sites, then the adsorption efficiency of the material is reduced” is an overinterpretation.
Response 3: Thanks for your comments. Based on your opinion, we have removed the overinterpretation (revised in manuscript, Line 125).
Comments 4: The t0 points in Figures 5A and 5B (for Fe@N co-doped BC) do not have corresponding values. This may affect the interpretation of the results and conclusions about the type of kinetics.
Response 4: Thanks for your careful review. Based on your opinion, we have listed the specific numbers below. In general, the authors only provided the degradation efficiency after the reaction in manuscript. Specific values will be presented through data sharing.
Comments 5: Please explain the low activity of the catalyst at pH 3.
Response 5: Thanks for your comments. Under the condition of acidity (pH = 3.0), the degradation efficiency of SMX decreased from 90.2% to 57.6%. Under acidic conditions, the increase in iron leaching would lead to a lower catalytic activity of Fe@N co-doped BC with fewer active sites (revised in manuscript, Line 165-168).
Comments 6: Please do not use the term "strong acidity" at pH 3. "acidity" is enough.
Response 6: Thanks for your comments. Based on your opinion, we have revised the relevant argument (revised in manuscript, Line 165-168).
Comments 7: I suggest moving Figures S2 and S4 from the supplement to the text.
Response 7: Thanks for your careful review. Based on your opinion, we have moved Figures S2 and S4 from the supplement to the text (revised in manuscript and supplementary material, Line 199, Line 320).
Comments 8: The results in Table S4 do not correlate with the information on the elution of Fe ions. Please explain this.
Response 8: Thanks for your comments. Fig. 8(d) displayed the mild leaching of Fe species in the redox process. However, the amounts of other elements also change during the reaction. As a result, the relative content of iron is slightly increased after the reaction. The data are not contradictory.
Comments 9: The formulas in Figure S4 will be more legible as structural patterns. The current form is an unnecessary eccentricity.
Response 9: Thanks for your careful review. Ball-and-stick models help readers to clearly understand where the pollutant molecules are attacked by active substances, so we choose more beautiful ball-and-stick models.
Comments 10: Figure 11 in its current form is difficult to read.
Response 10: Thank you very much for your careful review of our work. The surface of biochar is loaded with Fe3O4, the structure of the catalyst itself is relatively complex, and it is not easy to clearly and completely show the interaction between the PMS and the different active sites on the catalyst surface, so the figure needs to cover a lot of information.
Comments 11: In the first paragraph (3.2), please do not provide the SMX concentration.
Response 11: Thanks for your careful review. Based on your opinion, we have deleted the SMX concentration in the first paragraph (3.2) (revised in manuscript, Line 339-340).
Comments 12: For what purpose did the authors use the TOC analyzer?
Response 12: Thanks for your comments. After the degradation reaction, the substances in the system consists of the remaining SMX and intermediate products. The degradation efficiency of SMX was determined by high-performance liquid chromatography and the mineralization efficiency was determined by TOC analyzer. Therefore, in practical applications, it is necessary to consider the degradation efficiency and mineralization efficiency comprehensively to evaluate the integrity and effect of the treatment process.
Comments 13: The material used to synthesize the catalyst (BC) may be different. Please provide its elemental composition?
Response 13: Thanks for your careful review. Based on your opinion, we have provided the elemental composition of BC (revised in supplementary material, Table S4).
Table S4. Main elements of the obtained catalysts.
|
Samples |
C (at. %) |
N (at. %) |
O (at. %) |
Fe (at. %) |
|
BC |
65.3 |
0.2 |
24.0 |
5.0 |
|
Fresh Fe@N co-doped BC |
47.6 |
6.1 |
31.0 |
15.3 |
|
Used Fe@N co-doped BC |
39.6 |
5.3 |
35.0 |
20.1 |
Reviewer 2 Report
Comments and Suggestions for Authors
Generally, it is an intriguing manuscript, but its presentation is not clear in many places. Before being accepted for publication, the submitted manuscript needs a major revision based on the following comments:
· Why did the author choose the carbonization pyrolysis method to synthesize Fe @ N co-doped biochar when various other methods are available? Explain the reasoning behind this choice.
· In Figure 1, should the scale bar font of the SEM images be updated?
· In Figure 2, the figure caption contains multiple typographical errors. Should the font size in Figures 2(b) and 2(c) be increased?
· In the XRD pattern, should the crystalline size of the individual particles be calculated, and should the corresponding discussion be added to the revised manuscript?
· Could you check the entire manuscript for typographical errors, such as those found in the caption of Figure 3?
· Should the author check the entire manuscript for figure alignment issues, as the images and figure captions are not consistently placed? Should the author pay attention to this formatting?
Comments on the Quality of English LanguageTypographical errors are present throughout the manuscript. Authors are required to pay keen attention to this
Author Response
Dear Editor,
We would like to thank you very much for your time and consideration on our manuscript.
According to the reviewers’ comments, we have carefully revised the manuscript, which we hope meet with their approval. The following responses are inserted with each issue raised by the reviewers. The revisions are marked in red color in the revised manuscript.
Detailed responses to reviews’ comments
Reviewer #2
Generally, it is an intriguing manuscript, but its presentation is not clear in many places. Before being accepted for publication, the submitted manuscript needs a major revision based on the following comments:
Comments 1: Why did the author choose the carbonization pyrolysis method to synthesize Fe@N co-doped biochar when various other methods are available? Explain the reasoning behind this choice.
Response 1: Thank you very much for your careful review of our work. Based on previous research, Fe@N co-doped biochar could be generally prepared via one-pot calcination method and carbonization-pyrolysis method. Due to the simple operation process of the carbonization pyrolysis method, and the method has been proved to be reliable and stable, we choose the carbonization pyrolysis method (revised in manuscript, Line 326-327).
Comments 2: In Figure 1, should the scale bar font of the SEM images be updated?
Response 2: Thanks for your comments. Based on your opinion, we have re-labeled the clear scale in Figure 1 (revised in manuscript, Figure 1, Line 74).
Comments 3: In Figure 2, the figure caption contains multiple typographical errors. Should the font size in Figures 2(b) and 2(c) be increased?
Response 3: Thanks for your careful review. Based on your opinion, we have increased the font size in Figures 2(b) and 2(c) (revised in manuscript, Figure 2, Line 86).
Comments 4: In the XRD pattern, should the crystalline size of the individual particles be calculated, and should the corresponding discussion be added to the revised manuscript?
Response 4: Thanks for your careful review. XRD measurements were used to explore the crystallographic structures of the catalysts. It was demonstrated that Fe3O4 nanoparticles had been successfully loaded onto biochar surfaces. In the XRD pattern, we have obtained the most important information of Fe@N co-doped biochar, so we do not need to calculate the crystalline size of the individual particles. Besides, the discussion related to the crystalline size of the individual particles deviates from the main thrust of this manuscript.
Comments 5: Could you check the entire manuscript for typographical errors, such as those found in the caption of Figure 3?
Response 5: Thanks for your constructive suggestion. Based on your opinion, we have checked and corrected the typographical errors (revised in manuscript).
Comments 6: Should the author check the entire manuscript for figure alignment issues, as the images and figure captions are not consistently placed? Should the author pay attention to this formatting?
Response 6: Thanks for your constructive suggestion. Based on your opinion, we have carefully checked the entire manuscript for figure alignment issues, and made sure that the images are aligned with the position of the figure captions (revised in manuscript).
Reviewer 3 Report
Comments and Suggestions for Authors
Minor comments:
Line: 32-34: Explain more in detail about the benefits do persulfate-based advanced oxidation processes (PS-AOPs) provide over traditional techniques?
Line: 95-101: What distinguishes Fe@N co-doped BC from N-doped BC in terms of surface morphology, and why is this significant?
Line: 16-18, 369-373: How were the hydroxyl radicals (•OH) and sulfate radicals (SO4•−) recognized, and what portion do they performance in the degradation process?
Line: 42-44: Explain the Fe3+/Fe2+ redox cycle thought to be vital for producing ROSs during catalytic oxidation in detail.
Line: 414-420: During several cycles of degradation, and how did the authors estimate the reusability of Fe@N co-doped BC?
Line: 349-358: How does the degrading efficiency of SMX get affected by co-existing ions such as Cl− and HCO3−?
Line: 698-706: In what ways can the conclusions of this work help with the real-world implementation of Fe@N co-doped BC as a carbon catalyst?
Author Response
Dear Editor,
We would like to thank you very much for your time and consideration on our manuscript.
According to the reviewers’ comments, we have carefully revised the manuscript, which we hope meet with their approval. The following responses are inserted with each issue raised by the reviewers. The revisions are marked in red color in the revised manuscript.
Detailed responses to reviews’ comments are in the attachment.

Round 2
Reviewer 2 Report
Comments and Suggestions for Authors
In revised form, this manuscript justified all comments and has improved significantly to justify a publication